# Trimethylamine N-Oxide (TMAO) Plasma Levels in Patients with Different Stages of Chronic Kidney Disease

**DOI:** 10.3390/toxins17010015

**Published:** 2024-12-31

**Authors:** Marcia Ribeiro, Julie Ann Kemp, Ludmila Cardozo, Drielly Vargas, Marcelo Ribeiro-Alves, Peter Stenvinkel, Denise Mafra

**Affiliations:** 1Graduate Program in Biological Sciences—Physiology, Federal University of Rio de Janeiro (UFRJ), Rio de Janeiro 21941-630, Brazil; marciaribeiro@biof.ufrj.br (M.R.); dm@id.uff.br (D.M.); 2Graduate Program in Nutrition Sciences, Fluminense Federal University (UFF), Niterói 24220-900, Brazil; kempjulie@gmail.com (J.A.K.); ludmilacardozo@id.uff.br (L.C.); 3Division of Nephrology, Federal University of Rio de Janeiro (UFRJ), Rio de Janeiro 21941-630, Brazil; drielly.vargas@gmail.com; 4HIV/AIDS Clinical Research Center, National Institute of Infectology (INI/Fiocruz), Rio de Janeiro 21040-360, Brazil; mribalves@gmail.com; 5Division of Renal Medicine, Department of Clinical Science, Technology, and Intervention, Karolinska Institute, 141-86 Stockholm, Sweden

**Keywords:** chronic kidney disease, uremic toxins, dialysis, trimethylamine N-oxide

## Abstract

Background: In patients with chronic kidney disease (CKD), trimethylamine n-oxide (TMAO) accumulation exacerbates inflammation and contributes to oxidative stress. These complications are putatively linked to the development of cardiovascular diseases. Despite the known associations, the variation in TMAO plasma levels across different CKD stages and dialysis modalities remains underexplored. This study aimed to quantify TMAO plasma levels in different CKD stages and dialysis treatments. Methods: This cross-sectional study assessed TMAO plasma levels in non-dialysis CKD patients (ND), patients undergoing hemodialysis (HD), and peritoneal dialysis (PD). TMAO plasma levels were assessed by liquid chromatography coupled to triple mass spectrometry quadrupole. Results: In total, 15 ND patients [stages 3–5, glomerular filtration rate 41.4 mL/min/1.73 m^2^, 64 (IQR = 12.5) years, BMI 25.2 kg/m^2^, eight women]; 14 PD patients [57.5 (IQR = 8.5) years, BMI of 27.8 kg/m^2^, nine women]; and 34 HD patients [43.5 (IQR = 45.5) years, BMI of 24.4 kg/m^2^, nineteen women] were analyzed. ND patients had lower TMAO levels when compared to the HD (*p* < 0.0001) and PD patients (*p* = 0.001). There was no difference in TMAO levels between patients undergoing dialysis (*p* < 0.59). There was a negative correlation between TMAO and HDL plasma levels [rho = −0.380 (*p* < 0.004)], calcium [rho = −0.321 (*p* < 0.016)], and albumin [rho = −0.416 (*p* < 0.001)]. In addition, a positive correlation between TMAO and urea levels was observed [rho = 0.717 (*p* < 0.001)]. Conclusions: CKD stages impact TMAO levels since patients on non-dialysis treatment had lower levels than patients on HD and PD.

## 1. Introduction

Chronic kidney disease (CKD) is recognized by structural, electrolytic, histological, and functional abnormalities of the kidneys that persist for more than three months. It has a global impact, affecting approximately 10–12% of the world’s population [1]. According to current guidelines, there are five stages of CKD (according to the glomerular filtration rate—GFR) and three categories based on the level of albuminuria. CKD is a progressive and irreversible condition that commonly requires renal replacement therapies, such as hemodialysis (HD) and peritoneal dialysis (PD), in the last stage (CKD5) of the disease [2].

Several factors contribute to the accelerated progression of CKD. Among them, uremia has been the focus of many researchers today, characterized by an accumulation of solutes that the kidneys would eliminate under normal conditions [3]. In the setting of CKD, as GFR decreases, uremia becomes more significant. Despite the effectiveness of dialysis treatment, the removal of substances such as uremic toxins is impaired. Therefore, in patients undergoing dialysis, an accumulation of uremic toxins is commonly seen, which significantly contributes to cardiovascular outcomes [3,4].

According to The European Uremic Toxin Work Group (EUTox) classification, more than 100 uremic toxins have been identified, approximately 25% of which are highly bound to proteins and, therefore, are more inefficiently eliminated by dialysis [5,6]. Several protein-bound uremic toxins accumulated in CKD patients originate from gut microbiota and are by-products generated from the degradation of amino acids or other nutrients. Trimethylamine N-oxide (TMAO), a low molecular weight uremic toxin, is recognized for its cardiovascular effects and contributes to renal fibrosis [4]. TMAO is produced through the intestinal metabolism of specific dietary components, including choline, phosphatidylcholine, L-carnitine, and betaine. Intestinal bacteria metabolize these compounds to generate trimethylamine (TMA). Subsequently, TMA is transported to the liver, where it is oxidized into TMAO by flavin-containing monooxygenases (FMOs) [7,8]. Dietary sources of TMA primarily include animal-based foods such as red meat (beef, pork, lamb, veal, processed meat, and ham), egg yolks, and other products such as whole milk, yogurt, cream cheese, and butter [9].

Elevated TMAO plasma levels are closely associated with cardiovascular outcomes such as atherosclerosis, and some pro-atherosclerotic mechanisms of TMAO have been proposed [10,11,12,13]. TMAO can bind the protein kinase R-like endoplasmic reticulum kinase (PERK), promoting metabolic dysfunction, increased Ca^2+^ release [14], impaired nitric oxide production, and reduced vasorelaxation of the aortic endothelium [15]. TMAO activates NF-κB signaling with activation of the NLRP3 inflammasome and consequent increase in endothelial permeability and vascular inflammation [12,16]. Also, TMAO interferes with reverse cholesterol transport, reducing high-density lipoprotein (HDL) in the liver and contributing to dyslipidemia by altering cholesterol regulation [17].

Linking plasma TMAO levels with renal function is crucial, as studies show significantly elevated levels in this population compared to healthy individuals [18]. In healthy individuals, average TMAO concentrations are around 5.8 μM/L, but in end-stage CKD, these levels can surge up to 13 times higher [18]. This rise is often attributed to increased trimethylamine-producing gut bacteria, driven by intestinal dysbiosis that worsens as CKD progresses [19]. Given these findings, our study focused on quantifying plasma TMAO in patients at various CKD stages and dialysis treatments. This could provide further insight into disease progression and associated cardiovascular risk. This emphasizes the clinical relevance of monitoring TMAO in CKD management and its potential as a marker of disease progression.

## 2. Results

In total, 63 patients were recruited and analyzed: 15 non-dialysis (ND) CKD patients, 14 peritoneal dialysis (PD) patients, and 34 hemodialysis (HD) patients. Table 1 shows the general characteristics of the patients in the three groups; no significant differences were observed between them. Table 2 shows significant differences between some biochemical parameters.

Furthermore, regarding the lipid profile, we observed a significant difference in total cholesterol levels concerning the ND group with the PD (*p*-value = 0.05) and HD (*p*-value = 0.01) groups. Significant differences were observed in HDL levels concerning the ND patients with the PD (*p*-value = 0.0004) and PD with HD (*p*-value = 0.003) groups. Furthermore, high triglyceride levels were observed in PD patients compared to ND (*p*-value = 0.03) and HD groups (*p*-value = 0.01) (Figure 1A–C).

Table 3 shows the food intake analysis. The only difference was in fiber intake between the HD and PD groups.

Lower TMAO plasma levels were observed in ND patients compared to PD (*p*-value = 0.001) and HD (*p*-value = 0.0001) groups (Figure 2).

Figure 3 shows the correlogram with TMAO levels negatively correlated with HDL-C [rho = −0.380 (*p*-value = 0.004)], calcium [rho = −0.321 (*p* < 0.016)], and albumin [rho = −0.416 (*p*-value = 0.001)] plasma levels and positively with urea plasma levels [rho = 0.717 (*p*-value = 0.001)].

## 3. Discussion

Scientific evidence indicates that TMAO strongly contributes to the development of tubulointerstitial fibrosis and amplifies the expression of pro-fibrotic genes. In addition to the classic cardiovascular effects already elucidated in the literature, this reinforces the importance of monitoring TMAO levels in patients with CKD [8,20].

This study aimed to analyze the uremic toxin TMAO plasma levels in patients with CKD at different stages and treatments, including non-dialysis, HD, and PD. We observed that ND patients had lower TMAO levels when compared to HD and PD patients.

Consistent with the literature, our results confirm that the lower the kidney function, the higher the TMAO plasma levels [4]. As observed in our results, Andrikopoulos et al. demonstrated that kidney function is the leading influencer of TMAO circulating levels [21]. Additionally, the researchers reported that GFR can mediate ~20% of the increase in TMAO when associated with age. In corroboration, several studies have demonstrated a strong negative correlation between TMAO levels and eGFR [18,22,23]. A bidirectional causal pathway between high TMAO and eGFR in CKD has been suggested, and it is unclear which comes first.

HD patients present higher TMAO concentrations than healthy individuals and non-dialysis CKD patients [24,25,26]. Moreover, high plasma TMAO levels in HD patients have been proposed to occur due to renal impairment and gut dysbiosis. Still, they are also probably due to an enhancement in the production of the Flavin-Containing Monooxygenase 3 (FMO3) enzyme [23].

Under physiological conditions, without CKD, mammals excrete approximately 95% of the TMAO through glomerular filtration and tubular secretion by the kidneys [22]. Regarding this, Pelletier et al. demonstrated that four hours of hemodialysis significantly reduced the TMAO plasma levels, raising the hypothesis that HD patients may increase the production of TMAO through the liver enzyme FMO3 and prefer the TMAO production pathway over urea [23].

On the other hand, it has already been demonstrated that HD patients present a higher abundance of microbes possessing the gene CutC/CntA, allowing these microbes to produce the TMA from nutrients. Consequently, the TMAO plasma levels in these groups of patients increase. Reinforcing these, Holle et al. demonstrated that patients with CKD stage 5 presented higher concentrations of the gene CntA and plasma TMAO levels than other stages of the disease [27]. Despite this study, we confirm that dialysis patients have a significantly higher concentration of TMAO than non-dialysis CKD patients, corroborating the line of reasoning about kidney function and TMAO. However, we cannot demonstrate through which mechanism this happened.

Regarding patients with CKD undergoing PD, the literature indicates that they also have higher levels of TMAO compared to the general population due to impaired excretion [28]. In parallel, it is known that non-infectious factors can cause peritonitis in PD, such as peritoneal catheter implantation, high glucose levels, or advanced glycation end products (AGEs). Thus, inflammation can promote irreversible lesions in the peritoneal membrane since neutrophil infiltration induces monocyte chemotaxis through the upregulation of monocyte chemoattractant protein-1 (MCP-1/CCL2). This process is followed by lipopolysaccharide binding to Toll-like receptor 4, initiating numerous signaling pathways in peritoneal macrophages and mesothelial cells. Activation of the inflammatory cascade causes the death of mesothelial cells and activates fibroblasts [29,30]. Consequently, prolonged peritoneal inflammation leads to peritoneal fibrosis and ultrafiltration failure [28]. This scenario may be further aggravated when levels of TMAO are high. According to Zhang et al. [28], when the serum TMAO levels > 50 µM/mL, there is a greater predictive risk of acute peritonitis, mortality rates, and interruption of the PD modality.

The effectiveness of PD in removing TMAO compared to HD still needs to be determined. Although both PD and HD effectively remove uremic toxins from the blood, PD is performed more continuously and may retain residual renal function slightly more extended than HD [31]. Thus, TMAO clearance in patients undergoing PD could be more efficient, which would justify lower TMAO levels in individuals who rely on PD [31,32]. These data agree with the results of the present study, in which the median TMAO value in individuals with PD was lower than that in individuals with HD.

This study reports a significant negative correlation between TMAO and HDL-C. In agreement with these findings, Xiong et al. also found a robust negative correlation after conducting a prospective study with 112 patients with suspected atherosclerotic cardiovascular disease [33]. Furthermore, they reported that patients with hyperlipidemia had significantly higher levels of TMAO than patients without hyperlipidemia [33]. A Chinese study of 130 patients undergoing coronary angiography showed that hyperlipidemic patients had significantly higher plasma TMAO levels compared to those without hyperlipidemia. They also found a negative correlation between TMAO and HDL [34]. A case-control study with adult metabolic syndrome observed that individuals with high TMAO levels presented low HDL levels [35]. Another study also found an inverse association between TMAO and choline levels with plasma HDL-C levels and phospholipid concentrations and a direct association with methylation markers [36].

In this context, it is well established that higher levels of total cholesterol and Low-Density Lipoprotein (LDL-C) are associated with an increased risk of cardiovascular disease (CVD). In contrast, HDL-C levels within the appropriate range are considered protective [37]. However, several studies have demonstrated associations between TMAO and increased risk of cardiovascular events and all-cause mortality in patients with CKD [18,26,38]. TMAO can modulate cholesterol and sterol metabolism at several sites in vivo, increasing the risk of atherosclerosis [31].

There are various pro-atherosclerotic mechanisms of TMAO. First is reverse cholesterol transport reduction, which can occur by suppressing the intestinal microbiota and dietary supplementation with TMAO. Second, there is an increased surface expression of type A scavenger receptor and CD36 of macrophages, as well as the formation of foam cells. Third, in the liver, TMAO could reduce the expression of cytochrome P450 family 7 subfamily A member 1 (CYP7A1), a key enzyme in cholesterol metabolism, which is associated with reduced expression and synthesis of bile acids, decreased bile acid distribution, and increased atherosclerosis. However, whether these changes contribute to the reductions in reverse cholesterol transport is unclear [10].

In contrast, a recent longitudinal and observational study of 127 individuals with cardiovascular diseases revealed a positive association between HDL-C and TMAO. Thus, these authors pointed to hypertriglyceridemia, hyperglycemia, and carbohydrate intake as possible justifications since such factors tend to be associated with lower levels of HDL-C [38].

According to Obeid et al. [34], the role of HDL-C, phospholipid synthesis, and methyl donors in determining plasma TMAO concentrations is unknown. It is known that there are only a few modifiable factors known to affect plasma TMAO, such as choline intake in the recommended range. However, only 11–15% of dietary choline is converted to TMAO, and a 10-fold increase in dietary choline intake reflects a modest increase in plasma and urine TMAO and trimethylamine concentrations in animals [39,40]. Finally, further studies are needed to understand the biological basis of the correlation between TMAO and plasma HDL-C levels.

This study has some limitations that should be considered. First, we understand that a larger sample size would be interesting to represent a more global result, especially concerning the group of patients on peritoneal dialysis. In addition, the assessment of food intake to assess trimethylamine consumption could be important data to complement the results. However, it is essential to highlight that this study corroborates the understanding of the TMAO levels according to the progression of CKD, in addition to shedding light on the possibility of TMAO becoming a new treatment target in CKD.

## 4. Conclusions

In this cross-sectional study, we demonstrated that the stage of CKD affects TMAO plasma levels. Furthermore, we found a negative correlation between TMAO levels and HDL-C. Given the critical role of increased TMAO levels in cardiovascular outcomes, this study provides further evidence and knowledge on how these plasma levels are in patients with CKD at different stages and treatments, potentially generating new strategies targeting these levels.

## 5. Materials and Methods

### 5.1. Study Design and Patients

This is a cross-sectional analysis of baseline data from CKD patients undergoing hemodialysis (performing three dialysis sessions per week, lasting 4 h), continuous ambulatory peritoneal dialysis (CAPD), and non-dialysis patients (stage 3–5). This study had the following inclusion criteria: men and women aged between 18 and 75 years, patients on dialysis or CAPD for at least six months, or on conservative treatment (for non-dialysis patients) with a low-protein diet for at least six months (0.6 g/Kg/day). The following were not included: pregnant women, transplanted patients, liver diseases, autoimmune, infectious, cancer, acquired immunodeficiency syndrome, patients using catabolic drugs, pre-, pro-, or symbiotic supplements, or patients who had used antibiotics and/or anti-inflammatory medications in the last three months. The research project was approved by the Ethics Committee of the Faculty of Medicine/UFF (number 39904520.8.0000.5243). The patients provided written informed consent before participating in the study.

### 5.2. Sample Analysis

Blood samples were collected in the morning after an overnight fast using Vacutainer^®^ Franklin Lakes, NJ, USA tubes with ethylenediaminetetraacetic acid (EDTA) as an anticoagulant (1.0 mg/mL) and without anticoagulant. Plasma and serum were separated by centrifugation (3500 rpm, 15 min at 4 °C) and stored at −80 °C. Biochemical parameters, including albumin, glucose, parathormone, calcium, phosphorus, potassium, high-sensibility C-reactive protein (hsCRP), urea, and lipid profile, were measured using commercial kits from BioClin^®^, Belo Horizonte, Brazil according to the manufacturer’s instructions.

### 5.3. Trimethylamine N-Oxide Analysis

The liquid chromatography technique coupled to triple mass spectrometry quadrupole (HPLC-EM/EM) was used to measure the TMAO plasma levels. This analysis was performed by the company CEMSA^®^ (São Paulo, Brazil).

### 5.4. Food Intake and BMI Analysis

Food intake was assessed using the 3-day food record technique, covering two weekdays and one weekend day. Energy and macronutrient intake analyses were estimated using DietBox^®^ software, https://dietbox.me/pt-BR. Body mass index (BMI) was evaluated as weight (kilograms) divided by height squared (meters).

### 5.5. Statistical Analysis

Data were presented as medians with interquartile ranges (IQRs), representing the spread between the 75th and 25th percentiles, or as frequencies (percentages) for categorical data and comparison among groups (i.e., in non-dialysis, PD, and HD CKD patients) were tested by either Kruskal–Wallis (continuous-numerical variables) or chi-squared (discrete-nominal variables) tests. Skewed continuous-numerical variables underwent log transformation. Linear multiple fixed-effect models were used for plasma hs-CRP and TMAO level inferences. All models were adjusted for confounding variables (i.e., age, sex, and dialysis duration) wherever applicable. All other variables in the linear multiple fixed-effect models were held at their mean values or equal proportions to estimate marginal mean values for the groups. Contrasts were constructed from these estimated mean marginal effects. The Tukey Honest Significant Difference (HSD) method was used to correct *p*-values by the number of comparisons. Similarly, correlation analyses were conducted using Pearson’s coefficients after adjustments for the confounding variables. Statistical significance was determined at *p* ≤ 0.05, with all analyses conducted using R version 4.2.1.e.

## Figures and Tables

**Figure 1 toxins-17-00015-f001:**
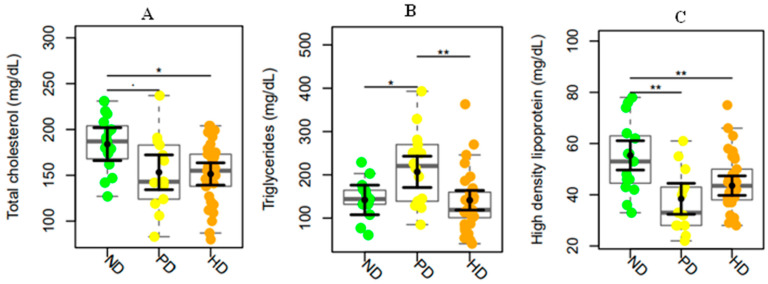
(**A**–**C**). Total cholesterol, triglycerides, and high-density lipoprotein plasma levels in non-dialysis (ND) patients, peritoneal dialysis (PD), and hemodialysis (HD) patients. The data distributions are represented in box and strip plots. In black, the center circles represent the mean marginal effects for each group estimated from a linear fixed-effects model adjusted for confounding variables (i.e., age and sex) by holding all other variables in the linear multiple fixed-effect models at their mean values or equal proportions. * *p*-values < 0.05, ** *p*-values < 0.01.

**Figure 2 toxins-17-00015-f002:**
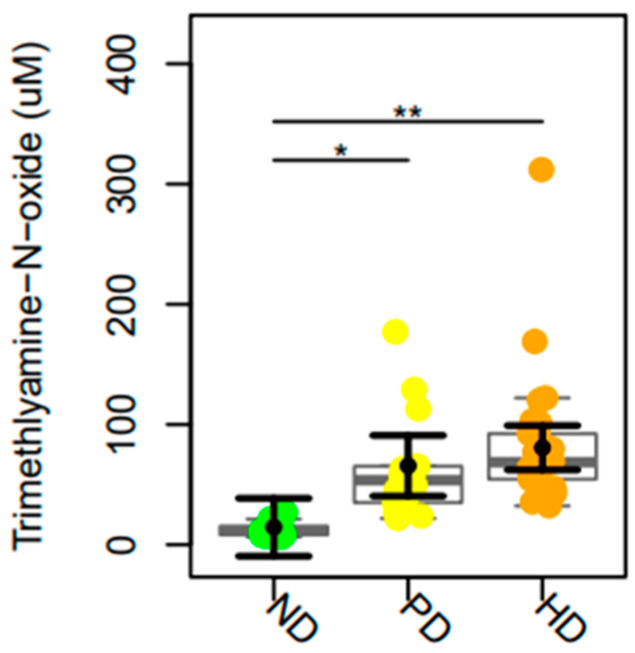
TMAO plasma levels in non-dialysis (ND) patients, peritoneal dialysis (PD), and hemodialysis (HD) patients. The data distributions are represented in box and strip plots. In black, the center circles represent the mean marginal effects for each group estimated from a linear fixed-effects model adjusted for confounding variables (i.e., age and sex) by holding all other variables in the linear multiple fixed-effect models at their mean values or equal proportions. Abbreviations: ND: non-dialysis patients; PD: peritoneal dialysis; HD: hemodialysis. * *p*-values < 0.05, ** *p*-values < 0.01.

**Figure 3 toxins-17-00015-f003:**
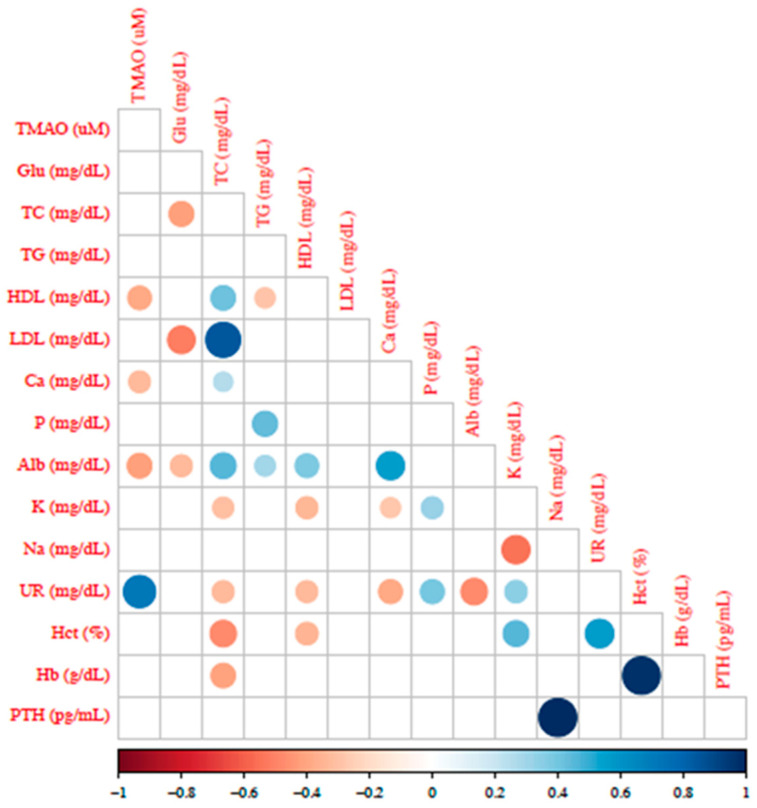
Correlogram of TMAO plasma levels and biochemical parameters in all patients with CKD. Only significant (*p* ≤ 0.05) positive (blue) and negative (red) correlations are shown in the graph. Abbreviations: glu: glucose; TC: total cholesterol; TG: triglycerides; HDL: high-density lipoprotein; LDL: low-density protein; Ca: calcium; P: phosphorus; Alb: albumin; K: potassium; Na: sodium; UR: urea; Hb: hemoglobin; OTH: parathormone.

**Table 1 toxins-17-00015-t001:** General characteristics of study participants.

Parameters	ND (N = 15)	PD (N = 14)	HD (N = 34)	*p*-Values
Sex (Male/Female)	7/8 (46.7/53.3%)	5/9 (35.7/64.3%)	15/19 (44.1/55.9%)	0.818
Age (years)	64 (12.5)	57.5 (8.5)	56 (16.2)	0.210
Time on dialysis (months)	-	24.5 (44.7)	43.5 (45.5)	0.091
eGFR (mL/min/1.72 m^2^)	44.0 (11.0)	-	-	-
BMI (Kg/m^2^)	25.2 (4.8)	27.8 (8.3)	24.4 (6.1)	0.371

Data expressed as either median and interquartile range (IQR) or absolute and relative (%) frequencies *p*-values estimated by nonparametric Kruskal–Wallis (continuous numerical variables) or chi-square tests (nominal/categorical variables). Abbreviations: ND: non-dialysis patients; HD: hemodialysis; PD: peritoneal dialysis; eGFR: estimated glomerular filtration rate; BMI: body index mass.

**Table 2 toxins-17-00015-t002:** Biochemical characteristics of study participants.

Parameters	ND (N = 15)	PD (N = 14)	HD (N = 34)
Glucose (mg/dL)	103.4 (70.0; 136.9) **^a.b^**	128.5 (92.8; 164.1) **^c^**	140.4 (114.2; 166.6)
Urea (mg/dL)	60.6 (46.9; 74.3) **^a.b^**	106.6 (92.1; 121.1) **^c^**	139.9 (129.9; 149.8)
Parathormone (pg/mL)	-	499 (153; 1153)	825 (497; 1152)
Hemoglobin (g/dL)	-	11.0 (10.0; 11.9)	10.0 (10.3; 11.5)
Calcium (mg/dL)	10.1 (9.7; 10.4) **^a.b^**	9.3 (8.9; 9.6)	8.7 (8.5.8.9)
Phosphorus (mg/dL)	4.1 (3.5; 4.7) **^a.b^**	5.4 (4.7; 6.0)	4.9 (4.5; 5.3)
Albumin (g/dL)	4.6 (4.4; 4.8) **^a.b^**	4.1 (3.9; 4.3) **^c^**	3.8 (3.7; 3.9)
Potassium (mg/dL)	4.7 (4.4; 5.0) **^b^**	4.8 (4.5; 5.2)	5.1 (4.9; 5.3)
hsCRP (mg/dL)	0.21 (0.11; 0.31)	0.14 (0.04; 0.25)	0.16 (0.08.0.23)
TMAO (µM)	11.4 (7.0) **^a.b^**	53.5 (29.3)	68.6 (37.1)

Data are presented as mean (confidence range—IC). *p*-values are estimated by the nonparametric Kruskal–Wallis test (continuous numerical variables). Letter **^a^** represents *p* ≤ 0.05 between ND and PD groups, **^b^** represents *p* ≤ 0.05 between ND and HD groups, and **^c^** represents *p* ≤ 0.05 between PD and HD groups. Abbreviations: ND: non-dialysis patients; HD: hemodialysis; PD: peritoneal dialysis; hsCRP: high-sensibility C-reactive protein.

**Table 3 toxins-17-00015-t003:** Food intake analysis of the study participants.

Parameters	ND (N = 15)	PD (N = 14)	HD (N = 34)
Energy (kcal/d)	1573.4 (1296; 1850)	1498 (1204; 1792)	1414 (1235; 1592)
Carbohydrates (g/d)	220 (185; 255)	192 (153; 231)	192 (168; 215)
Protein (g/d)	65 (52; 79)	77 (62; 92)	65.9 (56; 74)
Lipids (g/d)	50.5 (41.0; 60)	44.3 (33.8; 54.7)	41.7 (35.4; 48.1)
Fibers (g/d)	16.1 (13.2; 18.9)	17.8 (14.7; 20.8) **^a^**	12.3 (9.8; 14.8)

Data are presented as mean (confidence range—IC). *p*-values are estimated by nonparametric Kruskal–Wallis (continuous numerical variables). Letter **^a^** represents *p* ≤ 0.05 between HD and PD groups. Abbreviations: ND: non-dialysis patients; HD: hemodialysis; PD: peritoneal dialysis.

## Data Availability

Data are contained within the article.

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
