# Peer review of "Trimethylamine N-Oxide (TMAO) Plasma Levels in Patients with Different Stages of Chronic Kidney Disease"

_toxins, 2024, doi:10.3390/toxins17010015_

Round 1
Reviewer 1 Report
Comments and Suggestions for Authors
- The list of abbreviations is not correctly reported in the figures and
tables; I suggest correcting and adding all the acronyms used in the
captions of figures and tables to improve understanding. Furthermore, the
titles of the tables should also be improved and made homogeneous among all
the tables inserted. In table 2, the meaning of letters a, b and c is
unclear. I suggest improving the caption. The same in table 3 for the "a"
letter.
- I recommend replacing "normal condition" with "physiological conditions"
in the discussion section.
- I suggest reviewing the use of acronyms in the text; some acronyms are
not spelled out the first time they appear.
- In the statistical analysis section you wrote "Statistical significance
was determined at p-value < 0.05" (and not ≤) but in the results you
considered "statistically significant" the difference in total cholesterol
levels between the ND and PD groups with a p-value = 0.05. I recommend
revising this issue.
Author Response
The list of abbreviations is not correctly reported in the figures and tables; I suggest correcting and adding all the acronyms used in the captions of figures and tables to improve understanding. Furthermore, the titles of the tables should also be improved and made homogeneous among all the tables inserted.
Response: Thank you very much for your observation and suggestions. The appropriate changes have been made in the Ms and are highlighted.
In table 2, the meaning of letters a, b and c is unclear. I suggest improving the caption. The same in table 3 for the "a" letter.
Response: Thank you for your comment. We have made it clear.
- I recommend replacing "normal condition" with "physiological conditions" in the discussion section.
- I suggest reviewing the use of acronyms in the text; some acronyms are not spelled out the first time they appear.
Response: Thank you very much for this comment. We have made changes.
- In the statistical analysis section, you wrote "Statistical significance was determined at p-value < 0.05" (and not ≤) but in the results you considered "statistically significant" the difference in total cholesterol levels between the ND and PD groups with a p-value = 0.05. I recommend revising this issue.
Response: Thank you for this comment. It was our mistake. We have added the p-value ≤ 0.05.

Reviewer 2 Report
Comments and Suggestions for Authors
This manuscript examined the differences in uremic metabolite levels among CKD patients on HD, PD, and non-dialysis. Results suggest an elevation in TMAO levels in dialysis patients. Here are some questions:
1. One of the concerns is that it is unclear what the results really suggest. It is unclear whether the patients in the different groups had the same level of severity of disease, thus the results may indicate metabolite changes related to treatment but not severity. If they had varied degree of severity, can the TMAO difference be ascribed to treatment methods or disease severity itself?
2. The statistical analysis does not seem to account for any potential confounders of the patients, such as their GFR, sex, duration of disease.
3. It would be very helpful to measure the predialysis and postdialysis levels of TMAO to determine if the inefficiency in TMAO removal might be the contributor to the difference in TMAO.
4. Line 125-127 (Figure 4), were any of the correlations depending on treatment groups, i.e., were some of the correlations only seen in patients with a certain type of dialysis or non-dialysis?
5. Line 151-152. Not sure what results the authors were based on to conclude that TMAO levels were correlated with severity of disease.
Author Response
One of the concerns is that it is unclear what the results really suggest. It is unclear whether the patients in the different groups had the same level of severity of disease, thus the results may indicate metabolite changes related to treatment but not severity. If they had varied degree of severity, can the TMAO difference be ascribed to treatment methods or disease severity itself?
Response: Thank you very much for this comment. The various TMAO levels observed in CKD patients correlate with disease severity. As noted in our study and supported by existing literature, the more advanced the CKD, the higher the TMAO levels. Consequently, CKD patients undergoing HD and PD are more likely to have elevated TMAO levels due to renal failure, leading to greater disease severity.
The statistical analysis does not seem to account for any potential confounders of the patients, such as their GFR, sex, duration of disease.
Response: Thank you very much for this comment. As already described in the methods, all models were adjusted for confounding variables (e.g., age, sex, and dialysis duration).
It would be very helpful to measure the predialysis and postdialysis levels of TMAO to determine if the inefficiency in TMAO removal might be the contributor to the difference in TMAO.
Response: Thank you for this comment. We agree that this comparison is essential for future studies.
Line 125-127 (Figure 4), were any of the correlations depending on treatment groups, i.e., were some of the correlations only seen in patients with a certain type of dialysis or non-dialysis?
Response: Thank you very much for this comment. The correlations were performed across all patients participating in the study, including non-dialysis, HD, and PD patients.
Line 151-152. Not sure what results the authors were based on to conclude that TMAO levels were correlated with severity of disease.
Response: Thank you very much for this comment. As shown in Figure 2, non-dialysis patients had lower TMAO levels than HD and PD patients, corroborating the literature's findings that the greater the renal failure, the higher the TMAO levels.

Reviewer 3 Report
Comments and Suggestions for Authors
Finding the correlation between different parameters and diseases stages is very interesting and important topic. However, I don’t think that this paper presents novel and innovative research. Detailed comments are below:
1. The journal „Toxins” covers studies related to toxins and toxinology. I don’t think the presented manuscript is suitable for this journal.
2. Abstract: specific information about BMI, years etc. don’t need to be noted in abstract. However, there is lack of information about the aim of the study?
3. Figure 1. Flowchart – no patients were excluded, so it is enough if you write exclusion and inclusion criteria in materials and methods section. This figure is not necessary.
4. You should add table with the results showing TMAO plasma levels, because it was the main aim of the study, and you only shows the results in the figure.
5. In the discussion you write a lot about different publications about TMAO and kidney diseases. However, there are only few comparisons with your results, so the discussion should be reorganized.
6. There are no information about the measurements (only “standard laboratory method”)
7. What is the novelty of this study? You cited many papers dealing with TMAO plasma levels and kidney diseases. What is new in your paper?
Author Response
Abstract: specific information about BMI, years etc. don’t need to be noted in abstract. However, there is lack of information about the aim of the study?
Response: Thank you very much for this suggestion. The aim of the study was added in the abstract.
- Figure 1. Flowchart – no patients were excluded, so it is enough if you write exclusion and inclusion criteria in materials and methods section. This figure is not necessary.
Response: Thank you very much for this comment. The methods section highlighted the inclusion and exclusion criteria. We agree to remove Figure 1.
- You should add table with the results showing TMAO plasma levels, because it was the main aim of the study, and you only shows the results in the figure.
Response: Thank you very much for this comment. We have added the TMAO values in the Table 2.
In the discussion you write a lot about different publications about TMAO and kidney diseases. However, there are only few comparisons with your results, so the discussion should be reorganized.
Response: Thank you very much for this suggestion. We have restructured the discussion to present comparisons with our results. The appropriate changes have been highlighted in the discussion section.
- There are no information about the measurements (only “standard laboratory method”)
Response: Thank you very much. We have added more information.
What is the novelty of this study? You cited many papers dealing with TMAO plasma levels and kidney diseases. What is new in your paper?
Response: Thank you for this insightful comment. While our study does not propose an utterly novel question, it addresses a critical gap in the literature by providing robust evidence to support our hypothesis. Existing research has established associations between TMAO plasma levels and kidney diseases. Still, few studies have thoroughly examined the specific mechanisms linking TMAO with disease progression and cardiovascular risk in CKD. Our findings contribute to this understanding by emphasizing the importance of monitoring TMAO as a potential biomarker and therapeutic target in CKD management. This work strengthens the evidence base and highlights the translational relevance of incorporating TMAO assessment into clinical practice.

Round 2
Reviewer 2 Report
Comments and Suggestions for Authors
Thank you for responding to my questions.
The rationale and implication of the study may be better demonstrated to the audience if the authors can point out that the non-dialysis patients were those with less severe CKD and those with dialysis were more severe, thus the difference in TMAO between groups can be reflected as how TMAO was related to the severity of disease.
What about assessing the TMAO and GFR association?
Also, regarding the relationship between TMAO and lipid markers, adjusting for non-dialysis or dialysis in the statistical model may be necessary to determine if receiving this treatment may affect the associations as it may clean out the lipids and TMAO in different efficiencies.
Author Response
What about assessing the TMAO and GFR association?
Response: We did not observe a correlation between TMAO and GFR, likely due to the small number of patients in our study.
Regarding the relationship between TMAO and lipid markers, adjusting for non-dialysis or dialysis in the statistical model may be necessary to determine if receiving this treatment may affect the associations as it may clean out the lipids and TMAO in different efficiencies.
Response: Thank you for your insightful comment. We acknowledge the importance of adjusting for dialysis or non-dialysis treatment in the statistical model to account for their potential impact on lipid and TMAO levels. However, we did not perform this specific analysis in the current study. We recognize this limitation and will consider addressing it in future research to understand better the influence of different treatment modalities on these associations.

Reviewer 3 Report
Comments and Suggestions for Authors
Thanks a lot for your reply. The explanation of the study novelty is clear and reasonable. However, in the manuscript it is still not explained enough. Maybe you can add several sentences at the end of the introduction and/or in the conclusions?
Author Response
Maybe you can add several sentences at the end of the introduction and/or in the conclusions?
Response: Thank you for your suggestion. Due to the word limit constraints, adding more information to the introduction is not feasible. The conclusion reflects the findings of this simple cross-sectional study, which inherently has limitations. This conclusion is appropriate within the scope of the study.
